# The Limits of Tractable Marginalization

**Oliver Broadrick** [* 1]  **Sanyam Agarwal** [* 2]  **Guy Van den Broeck** [# 1]  **Markus Bläser** [# 2]

## Abstract

Marginalization – summing a function over all assignments to a subset of its inputs – is a fundamental computational problem with applications from probabilistic inference to formal verification. Despite its computational hardness in general, there exist many classes of functions (e.g., probabilistic models) for which marginalization remains tractable, and they can be commonly expressed by polynomial size arithmetic circuits computing multilinear polynomials. This raises the question, can *all* functions with polynomial time marginalization algorithms be succinctly expressed by such circuits? We give a negative answer, exhibiting simple functions with tractable marginalization yet no efficient representation by known models, assuming $\mathsf{FP} \neq \#\mathsf{P}$ (an assumption implied by $\mathsf{P} \neq \mathsf{NP}$). To this end, we identify a hierarchy of complexity classes corresponding to stronger forms of marginalization, all of which are efficiently computable on the known circuit models. We conclude with a completeness result, showing that whenever there is an efficient real RAM performing virtual evidence marginalization for a function, then there are small circuits for that function's multilinear representation.

## 1. Introduction

Marginalization is a fundamental computational problem. In machine learning for example, many architectures are designed around an instance of marginalization. In Bayesian learning (e.g., Bayesian neural networks), model parameters are marginalized to obtain a posterior distribution; variational autoencoders approximate a marginalization over their latent space; autoregressive and discrete diffusion models are trained to predict (conditional) marginal probabilities (Kingma et al., 2019; Bishop & Bishop, 2023). In the case of boolean valued functions, marginalization corresponds to the well known task of model counting, with applications from formal verification to cryptography (Gomes et al., 2021; Arteta et al., 2016).

Unfortunately, marginalization is a notoriously hard problem in general. As models become increasingly expressive-efficient, able to succinctly express larger classes of distributions, they often sacrifice the tractability of marginalization (Cooper, 1990; Roth, 1996). However, there is a vast body of work aimed at identifying classes of functions for which efficient marginalization remains tractable, including well known probabilistic models like hidden Markov models, determinantal point processes, and probabilistic circuits in general, as well as logical representation languages like d-DNNF (Rabiner & Juang, 1986; Kulesza et al., 2012; Choi et al., 2020; Darwiche, 2001). Moreover, a recent line of work has shown how known models with tractable marginalization can be commonly viewed as arithmetic circuits computing multilinear polynomials, resulting in a unified model that we formalize as *uniform finally multilinear arithmetic circuits (UFMACs)* (Zhang et al., 2020; 2021; Agarwal & Bläser, 2024; Broadrick et al., 2024b).

The fact that known functions with tractable marginalization can be expressed by polynomial size UFMACs begs the question. *Do all probability distributions over binary random variables with tractable marginalization have polynomial size (uniform) finally multilinear arithmetic circuits?* A positive answer to this question would show that the long line of work on tractable models has successfully arrived at a maximally expressive-efficient modeling language for tractable marginalization. We give a negative answer, suggesting the possibility of yet more expressive-efficient tractable models.

To rigorously answer this question, we define the tractability of marginalization for a function in terms of its polynomial time computability. Then, we observe that functions with UFMACs are actually tractable for more powerful forms of marginalization than standard variable marginalization. Specifically, we show that UFMACs support tractable summation over inputs of a given Hamming weight (Section 4)

---
[*,#]Equal contribution. [1]Department of Computer Science, University of California, Los Angeles, United States [2]Department of Computer Science, Saarland University, Saarbrücken, Germany. Correspondence to: Oliver Broadrick <odbroadrick@gmail.com>, Sanyam Agarwal <agarwal@cs.uni-saarland.de>.

*Proceedings of the $42^{nd}$ International Conference on Machine Learning*, Vancouver, Canada. PMLR 267, 2025. Copyright 2025 by the author(s).

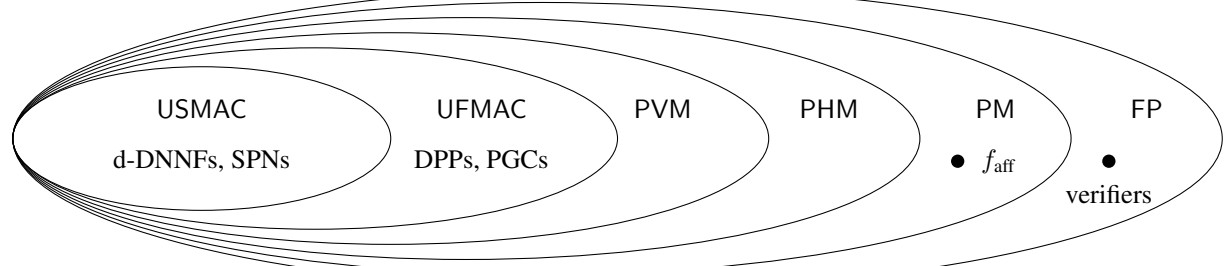

Figure 1. Relationships between complexity classes for tractable marginalization. UFMAC contains functions efficiently expressible using known tractable models including determinantal point processes (DPPs) and probabilistic generating circuits (PGCs) as well as the subclass USMAC containing for example, deterministic decomposable negation normal forms (d-DNNFs) and sum product networks (SPNs). PM is the set of all functions with polynomial time marginalization. PHM and PVM consist of functions that are tractable for Hamming weight marginalization and virtual evidence marginalization respectively (see Sections 4 and 5). We exhibit the function $f_{\text{aff}}$ which is contained in PM\PHM assuming FP $\neq$ #P. FP is the class of polynomial time computable functions, included for context.

and support marginalization in the presence of virtual evidence (Section 5). We show that the class of functions tractable for each of these queries form a hierarchy of inclusions, and we prove that certain inclusions are strict by exhibiting separating functions assuming standard complexity-theoretic conjectures (i.e., P $\neq$ NP): see Figure 1 for a visual summary. This reveals distributions that are tractable for marginalization but cannot be computed by a UFMAC, answering our central question.

While it turns out that not all distributions with tractable marginalization can be expressed by UFMACs, we then ask the corresponding question for the stronger forms of marginalization. We find that UFMACs are complete for tractable virtual evidence marginalization in the real RAM model of computation in the sense that if there is a polynomial time real RAM that performs virtual evidence marginalization for a function, then the function has UFMACs.

## 2. Tractable Arithmetic Circuits

Classical probabilistic models like hidden Markov models (Rabiner & Juang, 1986) and Chow-Liu trees (Chow & Liu, 1968) have efficient algorithms for marginalization based on sums and products of probabilities. For general graphical models this remains true, albeit with the caveat that the efficiency of the algorithm depends on the underlying graph structure. In the nineties, it was observed that such algorithms naturally correspond to the computation of certain multilinear polynomials (Castillo et al., 1995; Darwiche & Provan, 1997). Actually, a direct representation of such a polynomial can be exponentially more succinct than an equivalent graphical model (Roth & Samdani, 2009), and so recent tractable probabilistic models have focused on representing the underlying polynomial as an arithmetic circuit, a computation graph consisting of sums and products. Known models with tractable marginalization can be viewed in this

common language, including bounded-treewidth graphical models (Chow & Liu, 1968; Meila & Jordan, 2000; Rabiner & Juang, 1986; Koller, 2009), determinantal point processes (Kulesza et al., 2012), probabilistic sentential decision diagrams (Kisa et al., 2014), sum-product networks (Poon & Domingos, 2011), cutset networks (Rahman et al., 2014), characteristic circuits (Yu et al., 2024), and probabilistic generating circuits (Zhang et al., 2021; Harviainen et al., 2023; Bläser, 2023; Broadrick et al., 2024a). In this section, we formalize the unified tractable model as *finally multilinear arithmetic circuits* (UFMACs). A UFMAC expresses a function $f : \{0, 1\}^n \to \mathbb{Q}$ by its multilinear polynomial and represents the polynomial by an arithmetic circuit.

### 2.1. Multilinear Polynomials

Consider a function $f : \{0, 1\}^n \to \mathbb{Q}$. Among infinitely many polynomials that compute $f$ on $\{0, 1\}^n$, there is a unique *multilinear* polynomial (a fundamental object; for example, see O'Donnell (2014)). A polynomial is multilinear if its degree in every individual variable is at most one. The unique multilinear polynomial computing $f$ can be found by interpolation:

$$p(x_1, \ldots, x_n) = \sum_{S \subseteq [n]} f(v_S) \prod_{i \in S} x_i \prod_{i \notin S} (1 - x_i)$$

where $[n] = \{1, 2, \ldots, n\}$ and $v_S$ denotes the *characteristic vector* of the set $S$, that is, the element of $\{0, 1\}^n$ with $i$th entry 1 for $i \in S$ and $i$th entry 0 for $i \notin S$. It turns out that the tractability of marginalization for a function $f$ can be characterized to some extent in terms of the ability to evaluate its multilinear polynomial $p$ on elements of $\mathbb{Q}^n$ beyond just $\{0, 1\}^n$, as we will see in Sections 3 to 5.

## 2.2. Arithmetic Circuits

Arithmetic circuits are a thoroughly studied model for the computation of polynomials, an algebraic analogue to boolean circuits: see Shpilka et al. (2010) and Saptharishi (2015) for surveys.

**Definition 2.1** (Arithmetic Circuits). An arithmetic circuit over a field $\mathbb{F}$ in variables $X$ is a directed acyclic graph. A node with in-degree 0 is an *input* node and is labeled with an element in $\mathbb{F}$ or $X$. All other nodes are sum or product nodes, labeled accordingly by $+$ or $\times$. The node with out-degree 0 is the *output* node.[1]

Each node in an arithmetic circuit computes a polynomial: (i) each leaf computes the polynomial by which it is labeled, (ii) each sum node computes the sum of the polynomials computed by its children, and (iii) each product node computes the product of the polynomials computed by its children. The polynomial computed by a circuit is the polynomial computed by its output node.

Classes of arithmetic circuits that compute multilinear polynomials are themselves well studied (Nisan & Wigderson, 1996; Raz, 2009; Raz et al., 2008). A circuit is called *syntactically multilinear* if every product node has the property that its children mention disjoint sets of variables. An arithmetic circuit is called *(semantically) multilinear* if every node in the circuit computes a multilinear polynomial. However, there exist multilinear polynomials for which the only known polynomial size circuits do not conform to either of these (syntactic or semantic) multilinearity properties. One notable example is the determinant, a multilinear polynomial central to determinantal point processes (Kulesza et al., 2012), a popular tractable probabilistic model.

## 2.3. Uniform Finally Multilinear Arithmetic Circuits

In this work we consider a class of arithmetic circuits computing multilinear polynomials with (minimal) restrictions to guarantee efficient evaluation by a uniform algorithm, general enough to express known probabilistic models with tractable marginalization. In particular, while syntactically and semantically multilinear circuits force all internal nodes to compute multilinear polynomials, we allow intermediate nodes to accumulate higher degree in each variable. However, if intermediate degrees are allowed to grow arbitrarily, then the circuits can compute polynomials like $x^{2^n} + 2^{2^n}$ which cannot be efficiently evaluated (in polynomial time). We thus strike a balance, defining *finally multilinear* arithmetic circuits whose internal nodes compute polynomials of polynomially bounded degree, high enough to allow flexible algorithms like determinants, but low enough to guarantee polynomial time evaluation of the circuit at rational

points. Indeed, it is an open question whether polynomials like the determinant can be efficiently computed without this additional flexibility (Shpilka et al., 2010, Open Problem 3). In the following we deal with *families* of circuits $(C_n)_{n=1,2,\ldots} = (C_1, C_2, \ldots)$ with each $C_n$ a circuit, often using $(C_n)$ as shorthand for $(C_n)_{n=1,2,\ldots}$.

**Definition 2.2** (Finally Multilinear Circuits). The arithmetic circuit family $(C_n)$ is *finally multilinear* if for $n = 1, 2, \ldots$,

- $C_n$ computes a multilinear polynomial, and

- after replacing each of the constants in $C_n$, $c_1, \ldots, c_k \in \mathbb{F}$, by fresh variables $z_1, \ldots, z_k$, the polynomial computed by any internal node (in variables $x_1, \ldots, x_n, z_1, \ldots, z_k$) has total degree at most polynomial in $n$.

The only additional restriction needed to ensure that a circuit family can be efficiently evaluated at rational points is *uniformity*, which requires that there be an efficient method for obtaining a description of any circuit from the family. Uniformity rules out the unrealistic power of nonuniform circuit families which can, for example, compute undecidable languages.

**Definition 2.3** (Uniform Circuits). An arithmetic circuit family $(C_n)_{n=1,2,\ldots}$ is *uniform* if there exists a polynomial time Turing machine which on input $1^n$ outputs $C_n$ (in some reasonable fixed representation: see Appendix A.1).

We note that if an arithmetic circuit family $(C_n)$ is uniform, then $(C_n)$ is of polynomial size (since time bounds space). Definitions 2.2 and 2.3 together provide sufficient conditions for a circuit to be efficiently evaluated at rational points.

**Lemma 2.4.** *Let $(C_n)$ be a uniform finally multilinear arithmetic circuit family. Then there exists a polynomial time Turing machine which on input $x_1, x_2, \ldots, x_n \in \mathbb{Q}$ computes $C_n(x_1, x_2, \ldots, x_n)$.*

The efficient evaluation algorithm promised by Lemma 2.4 is simple: on input $n$ rationals, first obtain the circuit $C_n$ by uniformity, and then evaluate it in the natural way. We give the full proof that this is guaranteed to take polynomial time in Appendix A.2. The class of functions whose multilinear polynomial is computable by uniform finally multilinear arithmetic circuits constitutes the class of functions for which efficient marginalization algorithms are currently known, and thus the class of circuits central to this paper.

**Definition 2.5** (The Class UFMAC). A function family $(f_n)$ with[2] $f_n : \{0,1\}^n \to \mathbb{Q}$ is in UFMAC if there exists a uniform finally multilinear arithmetic circuit family computing the family of multilinear representations of $(f_n)$.

---

[1] We consider arithmetic circuits with a single output node, though in general they can have multiple.

[2] Note that fixing an encoding $\mathbb{Q} \to \{0,1\}^*$ allows functions $\{0,1\}^n \to \mathbb{Q}$ to be viewed as functions $\{0,1\}^n \to \{0,1\}^*$.

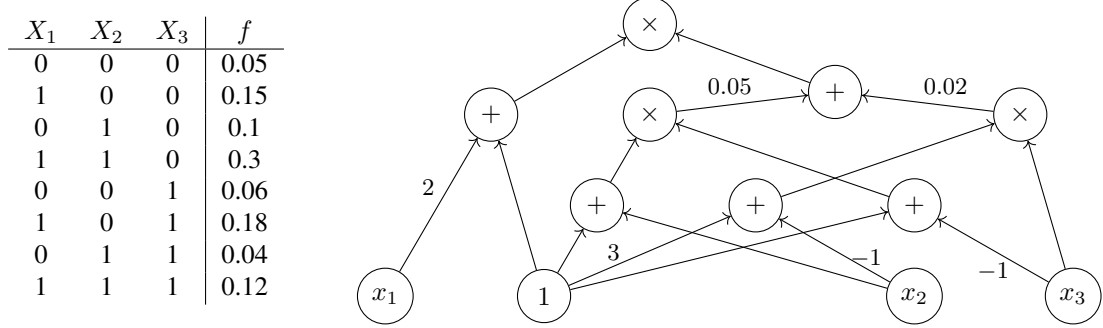

$$p(x_1, x_2, x_3) = .1x_1 + .05x_2 + .1x_1x_2 + .01x_3 - .07x_2x_3 + .02x_1x_3 - .14x_1x_2x_3 + .05$$

$$\bar{p}(x_1, \bar{x}_1, x_2, \bar{x}_2, x_3, \bar{x}_3) = 0.05\bar{x}_1\bar{x}_2\bar{x}_3 + 0.15x_1\bar{x}_2\bar{x}_3 + 0.1\bar{x}_1x_2\bar{x}_3 + 0.3x_1x_2\bar{x}_3 + 0.06\bar{x}_1\bar{x}_2x_3 + 0.18x_1\bar{x}_2x_3$$
$$+ 0.04\bar{x}_1x_2x_3 + 0.12x_1x_2x_3.$$

*Figure 2.* An example function $f$ (probability mass function over three binary random variables) and its multilinear polynomial $p$, network polynomial $\bar{p}$, and a multilinear arithmetic circuit computing $p$.

An important subclass of UFMAC is the analogously defined class of uniform *syntactically multilinear* arithmetic circuits (USMAC). A majority of known tractable models (both probabilistic and logical) belong to this subclass including probabilistic sentential decision diagrams (Kisa et al., 2014), sum-product networks (Poon & Domingos, 2011), d-DNNF (Darwiche, 2001), and more. Determinantal point processes are the best known example of a tractable model known to live in UFMAC but not known to live in USMAC.

In the next section we formalize the problem of marginalization, and show that efficient marginalization is easy for functions in UFMAC, giving a simple, unified view of existing marginalization algorithms.

**Complexity basics.** For notions from basic computational complexity theory, including complexity classes P, FP, NP, and #P, we use standard definitions and notation (Arora & Barak, 2009). We note that while the famous classes P and NP contain decision problems (functions of the form $f : \{0,1\}^* \to \{0,1\}$), FP and #P can be viewed as natural analogues for general functions, $f : \{0,1\}^* \to \{0,1\}^*$. In particular, FP contains all functions $f : \{0,1\}^* \to \{0,1\}^*$ computable in polynomial time. Moreover, FP $\neq$ #P is a well known conjecture in computational complexity theory and is a weaker assumption than P $\neq$ NP in the sense that P $\neq$ NP implies FP $\neq$ #P. Therefore, our results which assume FP $\neq$ #P also hold if P $\neq$ NP.

## 3. Tractable Marginalization

We define tractable marginalization in a natural way. For every function family $f = (f_n)$ with $f_n : \{0,1\}^n \to \mathbb{Q}$ we define a corresponding marginalization problem and then call $f$ tractable for marginalization if its marginalization problem is polynomial time computable.

**Definition 3.1** (MAR($f$)). Let $f = (f_n)$ with $f_n : \{0,1\}^n \to \mathbb{Q}$. The marginalization problem for $f$, denoted MAR($f$), gets input $m \in \{0,1,*\}^n$ and outputs $\sum_{x \in X_m} f_n(x)$ where

$$X_m = \{x \in \{0,1\}^n : x_i = m_i \text{ if } m_i \in \{0,1\}\}.$$

Intuitively, $m_i \in \{0,1\}$ means that the $i$th input is fixed ($x_i = m_i$), and $m_i = *$ means the $i$th input is marginalized, taking values freely in $\{0,1\}$.

*Example* 3.2 (Marginalization). Consider the function $f$ shown for $n = 3$ in Figure 2. Suppose we wish to marginalize over $X_2$ and $X_3$ with the evidence $X_1 = 0$. Specifically, on input $m = (0, *, *)$ to MAR($f$), we get

$$X_m = \{000, 010, 001, 011\}$$

$$\text{MAR}(f)(m) = \sum_{v \in X_m} p(v) = 0.05 + 0.1 + 0.06 + 0.04 = 0.25.$$

**Definition 3.3** (The Class PM). Let $f = (f_n)$ with $f_n : \{0,1\}^n \to \mathbb{Q}$. Then $f \in$ PM if MAR($f$) is polynomial time computable.

If $f \in$ PM, we say $f$ is tractable for variable marginalization. If $f$ has boolean codomain, then variable marginalization can be viewed as the task of model counting on $f$ after the substitution of variables given by any (partial) assignment. Of course, if a boolean function (family) $f$ is tractable for variable marginalization, then the function itself, $f(x)$, can be computed in polynomial time, by taking $m = x$.

**Proposition 3.4.** PM $\subseteq$ FP.

Moreover, it is not hard to see that a function being tractable for marginalization is a much stronger condition than the

function itself being tractable to compute. There exist many natural problems in FP\PM. For example, if FP $\neq$ #P, then every #P-complete problem provides a function in FP\PM.

**Proposition 3.5.** *If* FP $\neq$ #P, *then for every* #P-*complete function* $f$, *there is a function* $g \in$ FP\PM.

Examples of functions in FP\PM given by Proposition 3.5 include verifiers for NP-complete problems (e.g., satisfiability problems) as well as functions that do not correspond to NP-complete problems (e.g., a suitably defined indicator function for perfect matchings).

### 3.1. UFMAC **has Tractable Marginalization**

Having shown functions in FP\PM (assuming FP $\neq$ #P), we now consider functions in PM. Such functions include those expressible by the many existing tractable models that can be commonly viewed as uniform finally multilinear circuits. Efficient marginalization for functions in UFMAC follows from (i) the fact that FMACs can be evaluated at rational points in polynomial time and (ii) that to compute any marginalization of $f$ it suffices to evaluate the multilinear polynomial representation of $f$ at a single point. The former follows from Lemma 2.4. The elegant latter fact follows from a very simple argument using linearity of expectation given below for completeness, which has been observed before (Juma et al., 2009).

**Lemma 3.6.** *Let* $f : \{0,1\}^n \to \mathbb{Q}$ *have multilinear polynomial* $p$. *Then,*

$$\sum_{x \in \{0,1\}^n} f(x) = 2^n p(1/2, \ldots, 1/2).$$

*Proof.* We have

$$\begin{aligned}
\sum_{x \in \{0,1\}^n} p(x) &= 2^n \mathbb{E}_x p(x_1, \ldots, x_n) \\
&= 2^n p(\mathbb{E}_{x_1} x_1, \ldots, \mathbb{E}_{x_n} x_n) \\
&= 2^n p(1/2, \ldots, 1/2)
\end{aligned}$$

where the expectation is over the uniform distribution, and the middle equality follows by linearity of expectation. $\square$

(The above lemma can also be viewed as extracting the first Fourier coefficient of $f$, by first translating the domain $\{0,1\}$ to $\{-1,1\}$ by $x \mapsto 1 - 2x$ and evaluating at zero.) With the ability to compute sums of a multilinear polynomial with a single evaluation, we get that functions with uniform finally multilinear arithmetic circuits are tractable for marginalization.

**Proposition 3.7.** UFMAC $\subseteq$ PM.

*Proof.* Let $f \in$ UFMAC be computed by the UFMAC $(C_n)$. We need to provide a polynomial time Turing machine that computes MAR($f$). On input $m \in \{0, 1, *\}^n$, first obtain $C_n$ by uniformity of $f$. Then, for any $m_i \in \{0, 1\}$, replace any node in $C_n$ labeled by $x_i$ with $m_i$. Note that multilinearity of the polynomial computed is preserved under this substitution, and all remaining variables now need to be marginalized. By Lemma 3.6, it suffices to evaluate the remaining circuit with all inputs set to $1/2$, which takes polynomial time by Lemma 2.4. $\square$

## 4. Hamming Weight Marginalization

A key observation of our work is that UFMACs actually enable more powerful forms of marginalization. In this section we introduce the first such form, Hamming weight marginalization. The Hamming weight of a binary string $x \in \{0,1\}^n$ is the number of ones in it. Where standard marginalization asks for the sum of a function's values over all assignments with some subset of the inputs fixed, Hamming weight marginalization asks for the sum of its values only over the inputs of a given Hamming weight. In the case that the underlying function represents a probability distribution, Hamming weight marginalization enables conditioning on the Hamming weight of the binary random variables, or on the size of the random set, if the variables are interpreted as the characteristic vector of a set. Algorithms for this problem are known in some special cases of UFMACs, for example in 'structured' syntactically multilinear circuits (Vergari et al., 2021) and in determinantal point processes (Kulesza & Taskar, 2011; Calandriello et al., 2020).

**Definition 4.1** (HMAR($f$) and PHM). Let $f = (f_n)$ with $f_n : \{0,1\}^n \to \mathbb{Q}$. The Hamming weight marginalization problem for $f$, denoted HMAR($f$), takes input $m \in \{0, 1, *\}^n$ and $k \in \{0, 1, \ldots, n\}$ and gives output $\sum_{x \in X_{m,k}} f_n(x)$ where

$$X_{m,k} = X_m \cap \{x \in \{0,1\}^n : |x| = k\}.$$

Moreover, $f \in$ PHM if HMAR($f$) is polynomial time computable.

Intuitively, the input $m$ is the same as in MAR($f$), and the additional input $k$ further restricts the $x \in X_{m,k}$ being summed over to those with Hamming weight $|x| = k$. If $f \in$ PHM, we say $f$ is tractable for Hamming weight marginalization.

*Example* 4.2 (Hamming Weight Marginalization). Recall the distribution shown in Figure 2. Suppose we want to sum over $X_2$ and $X_3$, fixing $X_1 = 0$, all strings of Hamming weight $k = 1$. Specifically, we have input $m = 0**$, $k = 1$ to HMAR($f$). From Example 3.2, $X_m = \{000, 001, 010, 011\}$. So,

$$X_{m,1} = \{010, 001\}$$

$$\text{HMAR}(f)(m,1) = \sum_{v \in X_{m,1}} p(v) = 0.1 + 0.06 = 0.16$$

We first observe that a function tractable for Hamming weight marginalization, is also tractable for variable marginalization, since any variable marginalization query is the sum of $n+1$ Hamming weight marginalization queries for all $k \in \{0, 1, ..., n\}$.

**Proposition 4.3.** PHM $\subseteq$ PM.

We now show that UFMACs support tractable Hamming weight marginalization. To do so, we use a useful circuit transformation. For any multilinear polynomial $p(x_1, \ldots, x_n) = \sum_{S \subseteq [n]} c_S \prod_{i \in S} x_i$ with $c_S \in \mathbb{F}$, the *network* polynomial for $p$ is the multilinear polynomial

$$\bar{p}(x_1, \bar{x}_1, \ldots, x_n, \bar{x}_n) = \sum_{S \subseteq [n]} p(v_S) \prod_{i \in S} x_i \prod_{i \notin S} \bar{x}_i$$

where $v_S \in \{0,1\}^n$ is the characteristic vector of $S$. There is a polynomial time algorithm for transforming an arithmetic circuit computing a multilinear polynomial to a circuit computing its network form.

**Lemma 4.4** (Broadrick et al. (2024b)). *Given an arithmetic circuit of size $s$ computing polynomial $p$ in $n$ variables, an arithmetic circuit computing the network polynomial for $p$ can be constructed in time $O(sn)$.*

With the ability to efficiently transform UFMACs to compute their network polynomials, we can efficiently perform Hamming weight marginalization for UFMACs.

**Proposition 4.5.** UFMAC $\subseteq$ PHM.

*Proof.* Let $f \in$ UFMAC. We need to provide a polynomial time Turing machine that computes $\text{HMAR}(f)$. On input $m \in \{0, 1, *\}^n$, $k \in \{0, 1, \ldots, n\}$, first obtain a FMAC that computes $f_n$ by uniformity of $f$. Using Lemma 4.4, obtain a FMAC computing its network polynomial $\bar{p}(x, \bar{x})$. We will now evaluate $\bar{p}$ as follows. For each $i$, if $m_i = 0$, set $x_i = 0$ and $\bar{x}_i = 1$; if $m_i = 1$, set $x_i = t$ and $\bar{x}_i = 0$; if $m_i = *$, set $x_i = t$ and $\bar{x}_i = 1$. Note that $t$ is a single indeterminate/symbol used for all such $x_i$. Evaluating the circuit produces a univariate polynomial in $t$. The coefficient of the monomial of degree $k$ is the desired quantity. To see this, consider some monomial $p(v_S) \prod_{i \in S} x_i \prod_{i \notin S} \bar{x}_i$ in $\bar{p}$. The monomial evaluates to zero if $v_S$ does not agree with $m$ (i.e., if there is some $m_i = 1$ but $i \notin S$ or $m_i = 0$ but $i \in S$). Moreover, the degree of the monomial after the substitution is $|S|$, the Hamming weight of $v_S$, as needed. $\square$

*Example* 4.6 (Hamming Weight Marginalization). Recall Example 4.2. We summed over $X_2$ and $X_3$ with $X_1 = 0$, only strings of Hamming weight equal to 1. Following the proof of Proposition 4.5, we compute

$$\bar{p}(0, 1, t, 1, t, 1) = 0.05 + 0.1t + 0.06t + 0.04t^2,$$

and extract the coefficient of $t^k = t$. Hence, $\text{HMAR}(f)(m, 1) = 0.16$, as in Example 4.2.

## 4.1. Separating PHM from PM with CSPs

Our construction of a function that is tractable for marginals but #P-hard for Hamming queries exploits known dichotomy theorems for constraint satisfaction problems (CSPs). Constraint satisfaction problems concern conjunctions of *constraints*, each an application of some boolean relation $R \subseteq \{0,1\}^k$ to variables $x_1, \ldots, x_k$ (not necessarily distinct), i.e., $R(x_1, \ldots, x_k) = 1$ if and only if $(x_1, \ldots, x_k) \in R$. For some finite set $\Gamma$ of boolean relations (called a *constraint language*), a $\Gamma$-formula is a conjunction of constraints each an application of some relation in $\Gamma$. The constraint satisfaction problem CSP($\Gamma$) gives a $\Gamma$-formula $\phi$ as input and asks whether there is a satisfying assignment for $\phi$. The problem $k$-ONES($\Gamma$) gives a $\Gamma$-formula $\phi$ and an integer $k$ as input and asks whether there is a satisfying assignment for $\phi$ with exactly $k$ ones. The counting versions of these problems #CSP($\Gamma$) and #$k$-ONES($\Gamma$) ask for the number of satisfying assignments (in total and with $k$ ones respectively). Clearly, the complexity of these problems depends on the set of relations $\Gamma$.

There are elegant dichotomy theorems for these problems, stating that, depending on $\Gamma$, these problems are either polynomial time computable or #P-hard. We say a relation is affine if it is logically equivalent to a system of linear equations over GF(2), the finite field containing two elements.

**Theorem 4.7** (Creignou & Hermann (1996)). *If $\Gamma$ contains only affine relations, then #CSP($\Gamma$) is computable in polynomial time. Otherwise, #CSP($\Gamma$) is #P-complete.*

To the extent that #CSP($\Gamma$) is a CSP analogue to variable marginalization, #$k$-ONES($\Gamma$) is an analogue to Hamming weight marginalization. There is a similar dichotomy, but with fewer tractable cases, for #$k$-ONES($\Gamma$). We say a relation is width-$k$ affine if it is logically equivalent to a system of linear equations over GF(2) with each equation having at most $k$ variables with a nonzero coefficient.

**Theorem 4.8** (Creignou et al. (2010)). *If $\Gamma$ contains only width-2 affine relations, then #$k$-ONES($\Gamma$) is computable in polynomial time. Otherwise, #$k$-ONES($\Gamma$) is #P-complete.*

Leveraging these results from the CSP literature, we now construct a simple function for which standard variable marginalization is tractable but Hamming weight marginalization is #P-hard; in other words, we find a function tractable for marginals for which the existence of a UF-MAC would imply FP = #P. Consider the function $f_{\text{aff}} : \{0,1\}^{2n^3+n} \to \{0,1\}$ in variables $(x_i)_{i \in [n]}$,

$(y_{ijk})_{i,j,k\in[n]^3}$, and $(z_{ijk})_{i,j,k\in[n]^3}$ abbreviated to $\mathbf{x}, \mathbf{y}, \mathbf{z}$:

$$f_{\text{aff}}(\mathbf{x}, \mathbf{y}, \mathbf{z}) = \bigwedge_{i,j,k\in[n]^3} y_{ijk} \oplus x_i \oplus x_j \oplus x_k$$
$$\wedge \bigwedge_{i,j,k\in[n]^3} y_{ijk} \oplus z_{ijk}. \qquad (1)$$

We first observe that marginalization of $f_{\text{aff}}$ is computable in polynomial time, mirroring a tractable case of Theorem 4.7. After substituting in any partial assignment, $f_{\text{aff}}$ specifies a linear system over $GF(2)$. Performing Gaussian elimination reveals $k$ linearly independent equations. If $k \geq n$, then there are no satisfying assignments. If $k < n$, then there are $2^{n-k}$ satisfying assignments, as each affine equation (parity constraint) independently halves the number of solutions.

**Proposition 4.9.** $f_{\text{aff}} \in \mathsf{PM}$.

To see that Hamming weight marginalization on $f_{\text{aff}}$ is hard, we reduce from #$k$-ONES($\Gamma$) for a chosen $\Gamma$ consisting of only width-three affine relations. The reduction encodes a width-three formula as evidence (a partial assignment) to $f_{\text{aff}}$, maintaining a relationship between the Hamming weight of the original formula and the resulting query. Intuitively, this may work because of the additional width of $f_{\text{aff}}$, though we do not rule out the possibility of a lower-width function also working. In the reduction, the variables $y_{ijk}$ are used to 'turn on' and 'turn off' the width-3 equations in variables $x_i$ as needed. The variables $z_{ijk}$ and the constraints $y_{ijk} \oplus z_{ijk}$ are then used to 'balance' the Hamming weight of satisfying assignments since exactly one of $y_{ijk}$ and $z_{ijk}$ will be one.

**Theorem 4.10.** *HMAR*($f_{\text{aff}}$) *is* #P-*hard. That is,* $f_{\text{aff}} \notin$ PHM *unless* FP = #P.

*Proof.* Consider the following constraint language:

$$\Gamma = \{a \oplus b \oplus c\} = \{\{(0,0,1), (0,1,0), (1,0,0), (1,1,1)\}\}.$$

By Theorem 4.8, #$k$-ONES($\Gamma$) is #P-complete. We show the #P-hardness of HMAR($f_{\text{aff}}$) by providing a deterministic, polynomial time reduction from #$k$-ONES($\Gamma$). We get input a $\Gamma$-formula $\phi(x_1, \ldots, x_n)$ and an integer $k \in \{0, 1, \ldots, n\}$. We construct an 'evidence string' $m$ for HMAR($f$) as follows, consisting of an entry in $\{0, 1, *\}$ for each variable $x_i, y_{ijk}, z_{ijk}$. For any constraint $x_i \oplus x_j \oplus x_k = 1$ in $\phi$, set $y_{ijk} = 0$ and $z_{ijk} = 1$. All other variables are marginalized, i.e., their entry in $m$ is set to $*$. Then #$k$-ONES($\Gamma$)($\phi, k$) = HMAR($f_{\text{aff}}$)($m, k + n^3$). To see this, suppose $\phi(x) = 1$; we find the only $y$ and $z$ such that $f(\mathbf{x}, \mathbf{y}, \mathbf{z}) = 1$, and we then observe that $|\mathbf{x}, \mathbf{y}, \mathbf{z}| = |\mathbf{x}| + n^3$. Every constraint of $\phi$ is satisfied, and so every width-4 constraint in $f$ with $y_{ijk} = 0$ is satisfied. For constraints $x_i \oplus x_j \oplus x_k$ not in $\phi$, the corresponding width-4 clause in $f$ is satisfied by setting the free variable $y_{ijk}$ to

whichever value (0 or 1) is necessary. The values $z_{ijk}$ are then set to the opposite of the values of $y_{ijk}$ which satisfies the remaining width-2 clauses of $f$. For every $i, j, k \in [n]^3$ we have $z_{ijk} \neq y_{ijk}$, and so $|\mathbf{y}| + |\mathbf{z}| = n^3$. $\qquad \square$

Having shown that HMAR($f_{\text{aff}}$) is #P-hard, we have established that, unless FP = #P, the function $f_{\text{aff}}$ is not in UFMAC. In particular, this means that $f_{\text{aff}}$ cannot be efficiently represented by known tractable logical representation languages like d-DNNF or BDD, nor by tractable probabilistic models like SPNs or DPPs. This answers our central question negatively, showing that known models for tractable marginalization do not characterize the class of functions with tractable marginalization, suggesting the potential for yet more expressive-efficient tractable models. We remark that our separating example avoids the known efficient reductions from weighted to unweighted variants of counting problems, for example in weighted model counting (Chakraborty et al., 2015) and weighted CSP counting variants (Bulatov et al., 2012). We also note that Theorem 4.10 partially resolves an open question on the succinctness of languages in the knowledge compilation literature (Koriche et al., 2013) as described in Appendix D.

## 5. Virtual Evidence Marginalization

Interestingly, the computational problems we have studied so far can be characterized by the ability to evaluate the multilinear polynomial representation of the underlying function on increasingly large domains. To efficiently compute $f$ itself, one need only efficiently evaluate its multilinear polynomial $p$ on $\{0, 1\}^n$. To efficiently compute MAR($f$), it suffices to efficiently evaluate $p$ on $\{0, 1/2, 1\}^n$. For HMAR($f$), it suffices to evaluate $p$ on a set of the form $\cup_{i=1}^{n-2} (\{0, 1/2, 1, a_i\}^n)$ for distinct $a_i \in \mathbb{R}$ as shown below in Proposition 5.3. This begs the question: What power is afforded by the ability to evaluate the multilinear polynomial at all rational points? Nicely, this task corresponds naturally to probabilistic inference in the presence of *virtual evidence*.

Virtual evidence is a well known generalization of the standard notion of (hard) evidence (Bilmes, 2004). Where observing hard evidence means that the probabilities for some subset of outcomes in a distribution be 'zeroed out', virtual evidence allows the rescaling of the probabilities by arbitrary (nonnegative) weights. Given a probability distribution over variables $X_1, \ldots, X_n$, observing – conditioning on – the hard evidence that $X_i = 1$ can be viewed as scaling the probability for all outcomes with $X \neq 1$ by a factor of 0 (and then renormalizing the distribution). On the other hand, it is often necessary, for example in Bayesian belief updates, to multiply the probability of outcomes with $X_i \neq 1$ by factors other than 0.

We consider a general version of the problem of updat-

ing a distribution given virtual evidence, though multiple formulations are possible (Chan & Darwiche, 2005). Let $X_1, \ldots, X_n$ be random variables supported on $\{0, 1\}$ with joint probability mass function $p(x_1, \ldots, x_n)$. Given scaling factors $\alpha_1, \bar{\alpha}_1, \ldots, \alpha_n, \bar{\alpha}_n \in \mathbb{R}^{\geq 0}$ with either $\alpha_i > 0$ or $\bar{\alpha}_i > 0$ for each $i$, the mass function obtained after observing $(\alpha_1, \bar{\alpha}_1, \ldots, \alpha_n, \bar{\alpha}_n)$-virtual evidence is given by

$$p(x_1, \ldots, x_n) \prod_{i=1}^{n} (\alpha_i x_i + \bar{\alpha}_i (1 - x_i)) \qquad (2)$$

up to normalization. (Note that ignoring normalization for distributions in PM is not a computational issue, as normalization requires a single call to MAR$(f)$, namely with all coordinates marginalized.) We formulate the problem of virtual evidence marginalization as follows.

**Definition 5.1** (VMAR$(f)$ and PVM)**.** Let $f = (f_n)$ with $f_n : \{0, 1\}^n \to \mathbb{Q}$, and let $p = (p_n)$ be the family of multilinear polynomials computing $f$. The virtual evidence marginalization problem for $f$, denoted VMAR$(f)$, gets input $x_1, x_2, \ldots, x_n \in \mathbb{Q}$ and outputs $p(x_1, \ldots, x_n)$. Moreover, $f \in$ PVM if VMAR$(f)$ is polynomial time computable.

If $f \in$ PVM, we say $f$ is tractable for virtual evidence marginalization. That VMAR$(f)$ enables the incorporation of virtual evidence in the sense of Equation (2) follows by a straightforward application of Lemma 4.4 which we give in Appendix B. We immediately get that VMAR$(f)$ can be solved for UFMACs by simply evaluating (Lemma 2.4).

**Proposition 5.2.** UFMAC $\subseteq$ PVM.

The fact that functions in UFMAC permit efficient conditioning on virtual evidence generalizes known algorithms for special cases, including Bayesian networks with bounded treewidth (Bilmes, 2004) and certain syntactically multilinear circuits (Chan, 2017; Liu & Van den Broeck, 2021), with Proposition 5.2 giving a generalization of them. It is also straightforward to reduce Hamming weight marginalization to virtual evidence marginalization.

**Proposition 5.3.** PVM $\subseteq$ PHM.

*Proof.* Suppose VMAR$(f)$ is polynomial time computable. Consider the algorithm given in the proof of Proposition 4.5. This algorithm defines a univariate polynomial $q(t)$ in terms of the multivariate (multilinear) polynomial $p(x_1, \ldots, x_n)$ by setting each $x_i$ to either a constant or to $t$. The degree of $q(t)$ is at most $n$. Therefore, all the coefficients of $q(t)$ can be recovered by interpolation by evaluating at $n + 1$ distinct points using the polynomial time algorithm for VMAR$(f)$. Now, HMAR$(f)$ is simply the appropriate coefficient, as per the proof of Proposition 4.5. $\square$

Intuitively, this shows that despite Definition 5.1 containing no explicit summation, there is in some sense a hidden

marginalization problem in every VMAR query. On the one hand, multiplying the inputs by arbitrary rationals can be viewed as updating the distribution given virtual evidence, but on the other hand, we have observed (Lemma 3.6) that marginalization of $f$ can be reduced to evaluating the multilinear polynomial $p$ for $f$ at $1/2, \ldots, 1/2$. Therefore the evaluation of $p$ at arbitrary rational points $a_1, \ldots, a_n$ can be viewed as first applying virtual evidence corresponding to $2a_1, \ldots, 2a_n$ to $x_1, \ldots, x_n$ to get $2a_1 x_1, \ldots, 2a_n x_n$ and then evaluating at $x_1 = \ldots = x_n = 1/2$, i.e., marginalizing. In particular, to compute a marginal probability given some virtual evidence, a single VMAR$(f)$ query suffices.

Intuitively, VMAR$(f)$ captures the computational power afforded by a UFMAC, which raises the question: are UFMACs 'complete' for virtual evidence marginalization in the sense that any function $f$ with an efficient algorithm (Turing machine) for VMAR$(f)$ in fact has a UFMAC? That is, do we have UFMAC = PVM? This is the version of a question (restricted to multilinear polynomials) which has been asked before and appears challenging to answer (Koiran & Perifel, 2011). We are able, however, to give such a completeness result in the real RAM model of computation.

### 5.1. Completeness for Real RAMs

Given the apparent difficulty of answering this question for discrete computation models (i.e., Turing machines) and that the question naturally extends to real inputs, we consider the real RAM model of computation in which real numbers are stored in constant space and manipulated by constant time arithmetic. See Appendix C for a brief introduction to the model, though we refer the reader to Erickson et al. (2024) for a full presentation. First, we extend the notion of UFMACs to the real RAM model. We say a function family $f = (f_n)$ with $f : \{0, 1\}^n \to \mathbb{R}$ with multilinear representation $p = (p_n)$ has a real RAM UFMAC if there exists a polynomial time real RAM which on input $1^n$ outputs an arithmetic circuit $C_n$ computing $p_n$, in some fixed standard representation (see Appendix C.1). Note that we no longer require the bound on degree since intermediate values are always a single word in the real RAM computation, avoiding the problem of circuit evaluation by Turing machines. We also extend the definitions of MAR, HMAR, and VMAR in the natural way to the real RAM model. We then observe that if $f$ has a polynomial time real RAM computing VMAR$(f)$, then the $f$ has real RAM UFMACs, showing that UFMACs are in that sense *complete* for virtual evidence marginalization. The proof, given in Appendix C, observes that if the real RAM computes the polynomial correctly on all points, then there is some branch of the computation (i.e., ignoring comparisons) which computes it on all points, from which an arithmetic circuit can be recovered.

**Proposition 5.4.** *Let $f = (f_n)$ with $f_n : \{0, 1\}^n \to \mathbb{Q}$. If there exists a polynomial time real RAM computing*

*VMAR($f$), then there exists a real RAM UFMAC computing $f$.*

## 6. Related Work and Conclusion

We highlight some closely related research areas. While we focus on theoretical limits of marginalization, there is much work on leveraging theoretical progress in practice (Sladek et al., 2023; Loconte et al., 2023; 2024; Wang & Van den Broeck, 2025). Moreover, probabilistic models with tractable marginalization, in addition to quickly developing as performant probabilistic models in their own right (Liu et al., 2024a), also appear as integral components in recent proposals for control and alignment of deep generative models (Zhang et al., 2023; 2024; Liu et al., 2024b; Ahmed et al., 2022) as well as numerous other applications (Wedenig et al., 2024; Saad et al., 2021). Where we study exact marginalization, approximate methods form their own vast area (Hoffman et al., 2013; Liu et al., 2023). We considered functions of binary variables and note work on productive reductions to this setting (Garg et al., 2024; Cao et al., 2023). Given our use of affine relations, we observe that such parity constraints appear in other work on boolean knowledge representation both in algorithms and languages (Fargier & Marquis, 2008; Koriche et al., 2013; Fargier & Marquis, 2014; de Colnet & Mengel, 2021; Chakraborty et al., 2021).

In summary, despite known models with tractable marginalization being expressible as UFMACs, we show that there exist functions with tractable marginalization, yet no UFMAC assuming FP $\neq$ #P. On the other hand, UFMACs support the more powerful Hamming weight marginalization and virtual evidence marginalization, being complete for the latter in the real RAM model of computation.

We conclude with two open questions. First, is the inclusion PVM $\subseteq$ PHM strict? Second, observing that all marginalization algorithms in this paper are amenable to parallelization, do there exist sequential (P-hard) marginalization problems?

## Acknowledgements

This work was funded in part by the DARPA ANSR, CODORD, and SAFRON programs under awards FA8750-23-2-0004, HR00112590089, and HR00112530141, NSF grant IIS1943641, and gifts from Adobe Research, Cisco Research, and Amazon. Approved for public release; distribution is unlimited.

## Impact Statement

This paper presents work whose goal is to advance the field of Machine Learning. There are many potential societal consequences of our work, none which we feel must be specifically highlighted here.

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

# A. Uniform Finally Multilinear Arithmetic Circuits

## A.1. Note on Representation

The choice of representation is not important up to polynomial changes in length. For example and completeness, one can take the description following (Arora & Barak, 2009, Definition 6.5), substituting node types in the natural way, and indicating an edge not with zero or one but with a rational $a/b$ with $a, b \in \mathbb{Z}$ given in base two.

## A.2. Efficient Circuit Evaluation

We show that uniform finally multilinear arithmetic circuits can be evaluated at rational points in polynomial time. To do so, we first show that if a circuit family can be evaluated at integral points in polynomial time then it can be evaluated at rational points in polynomial time, a reduction which indeed holds for general arithmetic circuit families. (We also remark that multilinearity is not used in any of the proceeding arguments, and so, indeed, any uniform arithmetic circuit family with the degree bound of Definition 2.2 can be efficiently evaluated in the same way.)

**Lemma A.1.** *If there is a polynomial time Turing machine for evaluating the arithmetic circuit family $C = (C_n)_{n=1,2,\ldots}$ at integral points $x_1, \ldots, x_n \in \mathbb{Z}$, then there is a polynomial time Turing machine that evaluates $C$ at rational points $x_1, \ldots, x_n \in \mathbb{Q}$.*

*Proof.* Consider the input $\frac{a_1}{b_1}, \frac{a_2}{b_2}, \ldots, \frac{a_n}{b_n}$, with each $a_i, b_i \in \mathbb{Z}$. Let $D = \prod_{i=1}^{n} b_i$, a common denominator. Let $c_i = a_i \prod_{j \in [n] \setminus \{i\}} b_j \in \mathbb{Z}$, and so $\frac{a_i}{b_i} = \frac{c_i}{D}$. Denote by $p$ the polynomial computed by $C_n$ with the degree of $p$ being $d^*$. Then, using a new variable $t$, observe that

$$p\left(t\frac{c_1}{D}, \ldots, t\frac{c_n}{D}\right) = \sum_{d_1,\ldots,d_n \in [0,1,\ldots,d^*]^n} \alpha_{d_1,\ldots,d_n} \prod_{i=1}^{n} \left(t\frac{c_i}{D}\right)^{d_i} \tag{3}$$

$$= \sum_{k=0}^{n} \left(\frac{1}{D}\right)^k t^k \sum_{d_1,\ldots,d_n \in [0,1,\ldots,d^*]^n : \sum_{i=1}^{n} d_i = k} \alpha_{d_1,\ldots,d_n} \prod_{i=1}^{n} (c_i)^{d_i}.$$

where $\alpha_{d_1,\ldots,d_n} \in \mathbb{Q}$. In particular, consider the univariate polynomial in $t$,

$$f(t) = p\left(tc_1, \ldots, tc_n\right) = \sum_{d_1,\ldots,d_n \in [0,1,\ldots,d^*]^n} \alpha_{d_1,\ldots,d_n} \prod_{i=1}^{n} (tc_i)^{d_i}$$

$$= \sum_{k=0}^{n} t^k \sum_{d_1,\ldots,d_n \in [0,1,\ldots,d^*]^n : \sum_{i=1}^{n} d_i = k} \alpha_{d_1,\ldots,d_n} \prod_{i=1}^{n} (c_i)^{d_i}.$$

The polynomial $f(t)$ is of degree at most $n$ and thus can explicitly by interpolation given the evaluation of $p$ at $n + 1$ distinct (indeed, integral) points. Then, having $f(t)$, the value $p\left(\frac{c_1}{D}, \ldots, \frac{c_n}{D}\right) = p\left(\frac{a_1}{b_1}, \ldots, \frac{a_n}{b_n}\right)$ can be recovered, following Equation (3), by multiplying the coefficient of $t^k$ in $f(t)$ by $D^{-k}$ for $k = 0, 1, \ldots, n$ and summing. (Note that, while recovering $f(t)$ in this black-box manner suffices in general, $f(t)$ can be obtained more directly in the case of uniform circuits by evaluating the circuit "symbolically" according to Equation (3), requiring 1 rather than $n + 1$ evaluations.) $\square$

We now show that uniform finally multilinear arithmetic circuits can be efficiently evaluated at rational points, Lemma 2.4, restated here for convenience.

**Lemma 2.4.** *Let $C = (C_n)_{n=1,2,\ldots}$ be a uniform finally multilinear arithmetic circuit family. Then there exists a polynomial time Turing machine which on input $x_1, x_2, \ldots, x_n \in \mathbb{Q}$ computes $C_n(x_1, x_2, \ldots, x_n)$.*

*Proof.* By Lemma A.1, it suffices to show that $C$ can be evaluated at integral points. First, obtain the circuit $C_n$ in polynomial time by the uniformity of $C$. Next, we show that evaluating $C_n(x_1, \ldots, x_n)$ in the natural way takes polynomial time, owing to the polynomial bound on the degree of internal nodes.

Assume inputs $a_1, \ldots, a_n \in \mathbb{Z}$ are given in base two (with an additional sign bit). For $a \in \mathbb{Z}$, denote by $||a||$ the number of bits needed to write $a$ in this way (e.g., $||a|| \leq 2 + \log_2 a$). Then, for integers $a, b$, we have $||a + b|| \leq 1 + \max\{||a||, ||b||\}$ (Fact 1) and $||a \cdot b|| \leq ||a|| + ||b||$ (Fact 2), and, moreover, such additions and multiplications are polynomial time computable.

Consider the augmented circuit $C'_n$ obtained by replacing the constants of $C_n$ by fresh variables. We say a node is of degree $d$ if the polynomial it computes is of degree $d$ (in both the original and fresh variables). Let $B(d)$ be the maximum number of bits needed to write the integer computed by any node of degree $d$ in $C'_n$. We show by induction that $B(d) \leq (3d - 1)N$, where $N$ is the length of the input (i.e., $N \approx \sum_{i=1}^{n} ||a_i||$). This completes the proof since the additions and multiplications themselves are polynomial time computable in this bitwidth, and $d$ is bounded by a polynomial in $n \leq N$ by Definition 2.3. Note that by uniformity of $C$ there exists a (univariate) polynomial $q$ upper bounding the size of the representation of $C'_n$ by $q(n)$.

For the base case, we upper bound $B(1)$ by considering all nodes of degree 1 by cases (there are no nodes in $C'_n$ of degree zero). For input nodes $x_i$ in $C'_n$ which were already variables in $C_n$, the bitwidth is at most $N \leq q(N)$ since they are evaluated as $a_i$. For input nodes $z_i$ in $C'_n$ corresponding to the constants of $C_n$, the bitwidth is at most $q(n) \leq q(N)$ by uniformity of $C$. The nodes of degree one considered so far thus require at most $q(N)$ bits each. All other nodes must be the sum of two[3] degree one nodes and since there are at most $q(n) \leq q(N)$ nodes total in $C'_n$, and by Fact 1, we get $B(1) \leq 2q(N)$. Proceeding by induction, assume $B(k) \leq (3d - 1)q(N)$, and consider $B(k+1)$. Nodes of degree $k+1$ are obtained either (i) as a product of lower degree nodes or (ii) as a sum of nodes with degree at most $k + 1$. For case (i) and by Fact 2, the maximum resulting bitwidth is at most $\max_{i \in [k]} B(i) + B(k + 1 - i) \leq (3(k+1) - 2)q(N)$ where the inequality follows from the inductive hypothesis. For case (ii), since there are at most $q(n) \leq q(N)$ nodes in the circuit, we can have at most $q(N)$ many additions of degree $k + 1$ nodes. Hence, by Fact 1 we have $B(k + 1) \leq q(N) + (3(k+1) - 2)q(N) = (3(k + 1) - 1)q(N)$ as needed. $\qquad\square$

## B. Virtual Evidence

Consider a function $f = (f_n)$ with $f_n : \{0,1\}^n \to \mathbb{Q}$ with multilinear polynomial $p = (p_n)$. Here, we interpret the function as a probability mass function. We show how to use $\text{VMAR}(f)$ to observe virtual evidence on this distribution, in a way which allows multiple such observations of virtual evidence (by its commutativity) and so maintaining the ability to compute VMAR queries (and so HMAR and MAR queries) on the resulting distributions. Consider the following expression with $\alpha_i x_i$ and $\bar{\alpha}_i \bar{x}_i$ not both zero (i.e., $x_i \in \{0,1\}$, $\bar{x}_i := 1 - x_i$, and $\alpha_i, \bar{\alpha}_i$ not both zero as is guaranteed for valid virtual evidence scaling factors).

$$\left( \prod_{i=1}^{n} (\alpha_i x_i + \bar{\alpha}_i \bar{x}_i) \right) p \left( \frac{\alpha_1 x_1}{\alpha_1 x_1 + \bar{\alpha}_1 \bar{x}_1}, \ldots, \frac{\alpha_n x_n}{\alpha_n x_n + \bar{\alpha}_n \bar{x}_n} \right) \tag{4}$$

$$= \left( \prod_{i=1}^{n} (\alpha_i x_i + \bar{\alpha}_i \bar{x}_i) \right) \sum_{S \subseteq [n]} p(v_s) \prod_{i \in S} \frac{\alpha_i x_i}{\alpha_i x_i + \bar{\alpha}_i \bar{x}_i} \prod_{i \notin S} \left( 1 - \frac{\alpha_i x_i}{\alpha_i x_i + \bar{\alpha}_i \bar{x}_i} \right) \quad \text{by interpolation}$$

$$= \sum_{S \subseteq [n]} p(v_S) \prod_{i \in S} \alpha_i x_i \prod_{i \notin S} \bar{\alpha}_i \bar{x}_i \tag{5}$$

$$= \left( \prod_{i=1}^{n} (\alpha_i x_i + \bar{\alpha}_i \bar{x}_i) \right) \sum_{S \subseteq [n]} p(v_S) \prod_{i \in S} x_i \prod_{i \notin S} \bar{x}_i \quad \text{using } x_i \neq \bar{x}_i \text{ and } x^2 = x \text{ for } x \in \{0,1\}$$

$$= \bar{p}(x_1, \bar{x}_1, \ldots, x_n, \bar{x}_n) \prod_{i=1}^{n} (\alpha_i x_i + \bar{\alpha}_i \bar{x}_i) \quad \text{by definition}$$

$$= p(x_1, \ldots, x_n) \prod_{i=1}^{n} (\alpha_i x_i + \bar{\alpha}_i \bar{x}_i) \quad \text{by definition}$$

Thus evaluating the expression Equation (4) allows one to condition on virtual evidence given an algorithm for $\text{VMAR}(f)$. Note that Equation (5) can be written as $\sum_{S \subseteq [n]} p(v_S) \left( \prod_{i \in S} \alpha_i \prod_{i \notin S} \bar{\alpha}_i \right) \prod_{i \in S} x_i \prod_{i \notin S} \bar{x}_i$. By inspection, this already gives the desired effect of virtual evidence (i.e., scaling the probabilities of outcomes by the corresponding factor); we show the final three steps to demonstrate alignment with the definition given in Equation (2) (and so relying on $x_i, \bar{x}_i \in \{0,1\}$).

---

[3]Enforcing at most two children per node is straightforward in polynomial time (and so changing the size of the circuit by at most a polynomial factor).

# C. Real RAM

The real RAM is a standard computational model used extensively for modeling computations involving real numbers, widely popular in the computational geometry community; we refer to Preparata & Shamos (2012) for more details. Several variants of the model have been developed, and here we briefly describe a recent formalization provided by (Erickson et al., 2024), which has the advantage of describing *uniform* algorithms. Every real RAM algorithm takes as input a pair of vectors $(a, b) \in \mathbb{R}^n \times \mathbb{Z}^m$, and has a fixed bit size $w$ of every word. Essentially, a word is any integer in $\{0, ..., 2^w - 1\}$ which can be represented as a sequence of $w$ bits. A word size $w = \Omega(log(n + m))$ suffices, with the additional advantage of providing $O(1)$ access to input data. The machine contains two random access arrays $W[0...2^w - 1]$ and $R[0...2^w - 1]$, to store words and exact real values, respectively. There is a machine program counter that is incremented at each step, and at any point the machine executes the step indicated by this counter. The runtime of the machine is the number of instructions executed by the machine before halting. On the word array, we are allowed constant and memory assignments, comparisons between different word values, rounded arithmetic operations like addition, subtraction, multiplication, and division, and bitwise boolean operations. For the real array, we are allowed constant assignments to 0 or 1, memory assignments, comparison with 0, and exact arithmetic operations such as addition, subtraction, multiplication, and division. Note that here we consider real RAMs without the ability to perform square roots on real inputs. The major points of difference between the real array and the word array are: while we are allowed to cast integers as reals, we do not allow casting reals to integers (for example, by using floor function), testing whether a real register stores an integer value, or any access to the binary representation of the real numbers.

## C.1. Real RAM UFMACs

We say that $f = (f_n)$ with $f : \{0, 1\}^n \to \mathbb{R}$ has a real RAM UFMAC if there exists a polynomial time real RAM which on input $1^n$ outputs an arithmetic circuit $C_n$ computing $f_n$. The real RAM operates on the padded input $(1^{p(n)}, 1^{p(n)})$, where $p(n)$ is a polynomial bound on the size of $C_n$, i.e., at most the running time of the real RAM. The output is a circuit in some standard representation, with the gates, edge relations, and edge weights encoded in the memory array on output, using real registers to store edge weights of the circuit.

## C.2. Proof of Proposition 5.4

*Proof.* Consider any input for VMAR($f$) given to the real RAM as the tuple $\langle W, R \rangle$ where $W$ represents the word array (storing $w$-bit numbers), and $R$ represents the array of reals. The important thing to note is that the input points for the VMAR query are given to us in $R$, and $W$ is empty initially. Also, as mentioned in Appendix C above, the only real operation affected by the word array is when we cast integers to reals. However, this can be replicated purely on the real registers by creating the constant (by repeated multiplication and addition) on the reals, and then using that. Since, any word size is bounded by $2^w$, we can make the constant in $poly(w)$ time by repeated multiplication. Hence, any computation on the word array doesn't actually contribute to the final output. For the sake of completeness, we do note that since all words are $w$-bit integers, any computation involving them is a bit operation, and hence can be arithmetized with only a polynomial blowup. We now move to the main argument of the proof: showing how to convert any arithmetic operations on the reals to an arithmetic circuit computation in polynomial time. We are allowed four operations: addition, subtraction, multiplication, and division. Clearly, the first three easily give rise to an arithmetic circuit. Whenever we see a division operator, we use the famous result of Strassen (1973) to eliminate the division operator. Since, by assumption, the real RAM computes a multilinear polynomial in the end, we know that the degree is bounded by $n$, and hence to remove the division operator we only need to consider truncations up to degree $n$. Further, to avoid degree blowup on the intermediate nodes, we only need to consider homogenous parts up to degree $n$. This can be achieved with only a polynomial blowup in size (Lemma 5.2 in Saptharishi (2015)).

Finally, let us now look at the computation of the real RAM. First assume that the real RAM does not make any comparison operations. Then the computation of the real RAM can be turned into an arithmetic circuit computing some polynomial $q$, by the arguments above. Note, at each point we only compute the homogenous parts upto degree $n$. By the definition of virtual evidence, $q(x)$ coincides with $p(x)$ for all $x \in \mathbb{Q}^n$ where $p$ is the multilinear polynomial representation of $f$. Since $q$ and $p$ are polynomials, they also have to be the same as polynomials. Since $p$ is multilinear, the obtained circuit is finally multilinear by definition.

Now assume that the real RAM does make comparisons. We get some computation tree. At each leaf $u$, we compute a polynomial $q_u$. This polynomial coincides with $p$ for each input $x$ that takes the path to $u$. For each leaf $u$, the set of

all such $x$ form a semialgebraic set. Since all these sets together cover the whole $\mathbb{R}^n$, there is at least one $u$ such that the corresponding semialgebraic set contains the product $I_1 \times \ldots \times I_n$ of nonempty open intervals. On this product, $q_u(x) = p(x)$, and therefore, $q_u$ and $p$ are the same polynomials. $\qquad\square$

## D. Succinctness Corollary

In the literature on *knowledge compilation*, representation languages for boolean functions are compared in their tractability for various transformations and queries as well as in their succinctness. Marginalization corresponds to the query of counting in the boolean setting (up to partial assignments), and Theorem 4.10 settles an open question on the succinctness of languages with tractable counting. There are several well known languages that support polynomial time counting (e.g., OBDD, SDD), but most of them are special cases of (and so not more succinct than) d-DNNF. Moreover, d-DNNF straightforwardly translates (in linear time) into syntactically multilinear arithmetic circuits, thus falling within UFMAC. However, Koriche et al. (2013) introduced the representation language Extended Affine Decision Tree (EADT) which supports polynomial time counting, and they left open the succinctness relationship between d-DNNF and EADT. While we refer the reader to Koriche et al. (2013) for full definitions, it is not hard to see that our function $f_{\text{aff}}$ can be represented efficiently in EADT (in fact even in the much less succinct though incomplete language AFF (Fargier & Marquis, 2008)). We are therefore able to partially resolve the succinctness relationship between d-DNNF and EADT.

For a formula $\phi$ in some representation language $L$, let $|\phi|$ denote the size of $\phi$ (say, the number of bits in some reasonable fixed representation). For representation languages $L_1$, $L_2$, we say $L_1$ is at least as succinct as $L_2$ (denoted $L_1 \leq_s L_2$) if there is a polynomial $p$ such that for any $\phi_2 \in L_2$, there exists a $\phi_1 \in L_1$ with $\phi_1 \equiv \phi_2$ such that $|\phi_1| \leq p(|\phi_2|)$. Here $\phi_1 \equiv \phi_2$ means that $\phi_1$ and $\phi_2$ represent the same boolean function.

**Corollary D.1.** *Unless* $\mathsf{PH} = \Sigma_2$, *d-DNNF* $\not\leq_s$ *EADT.*

*Proof.* Let $f_{\text{aff}}$ be as defined in Equation (1), and so $\text{HMAR}(f_{\text{aff}})$ is #P-hard by Theorem 4.10. Note that $f_{\text{aff}}$ can be represented in polynomial size in either AFF or EADT. But, if $f_{\text{aff}}$ can be represented by a polynomial size d-DNNF, then by the algorithm of Proposition 4.5 we have $\text{HMAR}(f_{\text{aff}}) \in \mathsf{P}\backslash\mathsf{poly}$. In particular, this gives $\#\mathsf{P} \subseteq \mathsf{FP}\backslash\mathsf{poly}$ and so $\mathsf{NP} \subseteq \mathsf{P}\backslash\mathsf{poly}$, collapsing the polynomial hierarchy to $\Sigma_2$ by the Karp-Lipton Theorem (Karp & Lipton, 1980). $\qquad\square$

