# OpenReview forum: "The Limits of Tractable Marginalization"
_ICML.cc/2025/Conference — ICML 2025 poster_

### Official Review · Reviewer_WSxE · 2025-03-02

**Overall Recommendation:** 3

**Summary:**

This paper defines three computation classes inclusing PM for tractable marginalization, PHM for Hamming marginalization and PVM for virtual evidence marginalization. The authors manage to show that class UFMAC is in all three classes mentioned before. Further, the authors show that PVM \subseteq PHM \subseteq PM. With the two conclusions above, the authors manage to provide a hierarchy of complexity class between UFMAC and FP.

**Claims And Evidence:**

The authors provide proves for all their claims.

**Essential References Not Discussed:**

N/A

**Experimental Designs Or Analyses:**

N/A

**Methods And Evaluation Criteria:**

N/A

**Other Comments Or Suggestions:**

N/A

**Other Strengths And Weaknesses:**

I think the paper is nicely written, and the result is clear and easy to understand. The three ways of marginalization are also interesting and intuitive. However, it is unclear how the paper relates to the theme of this conference. It seems to me the paper fits better for a conference like CCC. I wonder if the authors could explain more about why complexity classes relate to marginalization, which is interesting for people who work in learning theory or even machine learning.

**Questions For Authors:**

I wonder if there are any possible implications or applications of the result in machine learning or learning theory. I think it would be helpful if the authors could give some explanation on why PVM, PHM, and PM are important complexity classes.

**Relation To Broader Scientific Literature:**

N/A

**Theoretical Claims:**

N/A

---

> ### Author Rebuttal · Authors · 2025-03-31
>
> Thank you for your considered review.
>
> On our paper’s relation to the conference theme, we recall that ICML has recently featured other papers entirely focused on the theory of tractable marginalization [0, ICML 2023; 1, ICML 2024] (with similarly focused papers appearing at closely related conferences [8,9,10]). Moreover, we emphasize the fruitful tradition of a close connection between the theory and practice of tractable models, with theoretical insights inspiring practical breakthroughs. For example, [4, ICML 2024] leveraged theoretical expressivity theorems to inform the design of state-of-the-art tractable models (with similar examples appearing at closely related conferences [5,6,7]). More generally, we remark that marginalization is an essential task (i.e., computing the normalizing constant or partition function) for any probabilistic model; the question of tractable marginalization is therefore as old as probabilistic reasoning as a field.
>
> As for appeal to the broader community, we note that tractable models, in addition to quickly developing as performant probabilistic models in their own right [3], also appear as integral components in recent proposals for control and alignment of deep generative models [11,12,13] as well as numerous other applications [2,14].
>
> We therefore view our work as fundamental and relevant to ICML, developing our understanding of tractable models by formalizing (and making progress in answering) the most basic questions about when marginalization is and isn’t computationally tractable. As reviewer jvgm writes, our work “substantially enhances our understanding of tractable marginalization” while managing to remain “neat,” “precise,” and “elegant,” at times even “refreshingly simple”. We hope this further context is helpful in addressing your questions and concerns, and we welcome any further discussion.
>
> [0] Bläser. Not all strongly Rayleigh distributions have small probabilistic generating circuits. ICML 2023.
>
> [1] Agarwal et al. Probabilistic Generating Circuits - Demystified. ICML 2024.
>
> [2] Wedenig et al. Exact Soft Analytical Side-Channel Attacks using Tractable Circuits. ICML 2024.
>
> [3] Liu et al. Scaling Tractable Probabilistic Circuits: A Systems Perspective. ICML 2024.
>
> [4] Zhang et al. Probabilistic Generating Circuits. ICML 2021.
>
> [5] Loconte et al. Subtractive Mixture Models via Squaring: Representation and Learning. ICLR 2024.
>
> [6] Wang et al. On relationship between monotone and squared circuits. AAAI 2025.
>
> [7] Loconte et al. Sum of Squares Circuits. AAAI 2025.
>
> [8] Martens et al. On the expressive-efficiency of sum-product networks. NeurIPS 2013.
>
> [9] Wang et al. A Compositional Atlas of Algebraic Circuits. NeurIPS 2024.
>
> [10] Vergari et al. A Compositional Atlas of Tractable Circuit Operations for Probabilistic Inference. NeurIPS 2021.
>
> [11] Zhang et al. Tractable Control for Autoregressive Language Generation. NeurIPS 2023.
>
> [12] Zhang et al. Adaptable Logical Control for Large Language Models. NeurIPS 2024.
>
> [13] Liu et al. Image Inpainting via Tractable Steering of Diffusion Models. ICLR 2024.
>
> [14] Saad et al. SPPL: Probabilistic Programming with Fast Exact Symbolic Inference. PLDI 2021.

---

> > ### Comment · Reviewer_WSxE · 2025-04-01
> >
> > Thanks to the authors for the detailed mention of related work. I am less familiar with tractable marginalization, and this answer provided by the authors is a good tutorial. I understand the conference has a page limit, but it would be nice for the authors to include this in the related work section in the final version for general audiences if the paper is accepted. I raised my score accordingly.

---

> > > ### Author Response · Authors · 2025-04-02
> > >
> > > Thank you for your prompt response. We will incorporate further context along these lines into the final version of our paper. If you have any remaining questions or concerns, we welcome further discussion.

---

### Official Review · Reviewer_DbFA · 2025-03-09

**Overall Recommendation:** 4

**Summary:**

The paper "The Limits of Tractable Marginalization" explores the computational complexity of marginalization, a fundamental operation in probabilistic inference and formal verification.

The authors focus on the relationship between functions with tractable marginalization and their representation using uniform finally multilinear arithmetic circuits (UFMACs)—a class encompassing known probabilistic models like hidden Markov models, determinantal point processes, and sum-product networks.

They show that while all known tractable models can be captured by UFMACs, not all functions with efficient marginalization necessarily admit such circuit representations.

Beyond standard variable marginalization, the authors examine two stronger forms of marginalization: Hamming weight marginalization and virtual evidence marginalization.

**Claims And Evidence:**

### **Claim 1: Not all functions with polynomial-time marginalization (PM) can be efficiently represented by uniform finally multilinear arithmetic circuits (UFMACs).**
   - **Evidence:** The paper constructs a function \( f_{\text{aff}} \) that has tractable marginalization but does not belong to UFMAC, assuming **FP ≠ #P**. The function \( f_{\text{aff}} \) is defined using **constraint satisfaction problems (CSPs)** with parity constraints.

---

### **Claim 2: UFMACs allow more powerful forms of marginalization than standard variable marginalization, such as Hamming weight marginalization.**
   - **Evidence:** The authors show that any function in **UFMAC** can efficiently compute **Hamming weight marginalization (HMAR)** queries using a transformation to **network polynomials**. They prove that for any function \( f \) represented as a UFMAC, its Hamming weight marginalization can be computed via polynomial-time interpolation.

---

### **Claim 3: There exists a strict separation between PM and PHM (assuming FP ≠ #P).**
   - **Evidence:** The function \( f_{\text{aff}} \) is used as a **separating example**. It is shown to be in PM (tractable for variable marginalization) but **not** in PHM, since computing Hamming weight marginalization on \( f_{\text{aff}} \) is #P-hard.

---

### **Claim 4: UFMACs allow efficient computation of virtual evidence marginalization (VMAR).**
   - **Evidence:** The paper proves that any function in UFMAC can efficiently compute virtual evidence marginalization via standard polynomial evaluation.


---

### **Claim 5: UFMACs are **complete** for virtual evidence marginalization in the real RAM model.**
   - **Evidence:** The paper proves that if a function has a polynomial-time **real RAM** algorithm for virtual evidence marginalization, then it has an efficient UFMAC representation.

---

### **Claim 6: The class of functions tractable for virtual evidence marginalization (PVM) is at least as large as the class of functions tractable for Hamming weight marginalization (PHM).**
   - **Evidence:** The authors prove **PVM ⊆ PHM** by showing that Hamming weight marginalization can be efficiently computed via virtual evidence marginalization.

**Essential References Not Discussed:**

NA.

**Experimental Designs Or Analyses:**

NA.
No experiments are conducted in this paper.

**Methods And Evaluation Criteria:**

This paper is purely theoretical, so no benchmark data is proposed and no empirical evaluation is conducted.

**Other Comments Or Suggestions:**

NA.

**Other Strengths And Weaknesses:**

NA

**Questions For Authors:**

NA

**Relation To Broader Scientific Literature:**

This paper addressed an important problem of computing marginalization of probabilistic models.

**Theoretical Claims:**

Yes. I have checked the proof  of theoretical claims:


- Claim 1: Not all functions with polynomial-time marginalization (PM) can be efficiently represented by uniform finally multilinear arithmetic circuits (UFMACs).
- Claim 2: UFMACs allow more powerful forms of marginalization than standard variable marginalization, such as Hamming weight marginalization.
- Claim 3: There exists a strict separation between PM and PHM (assuming FP ≠ #P).
- Claim 4: UFMACs allow efficient computation of virtual evidence marginalization (VMAR).
- Claim 5: UFMACs are **complete** for virtual evidence marginalization in the real RAM model.
- Claim 6: The class of functions tractable for virtual evidence marginalization (PVM) is at least as large as the class of functions tractable for Hamming weight marginalization (PHM).

---

> ### Author Rebuttal · Authors · 2025-03-31
>
> Thank you for your considered review.
>
> We see no questions or concerns in your review needing response. We nonetheless welcome any further discussion, e.g., as prompted by the other reviews.

---

### Official Review · Reviewer_PYPx · 2025-03-17

**Overall Recommendation:** 3

**Summary:**

This paper tackles the problem of characterizing the class of functions on which marginalization can be performed in polynomial time. Previous work describes a construction with polynomial size arithmetic circuits computing multilinear polynomials. The authors demonstrate that this construction is incomplete and produce a broader class of functions with polynomial marginalization.

**Claims And Evidence:**

Claims are supported by propositions, lemmas, theorems, etc.

**Essential References Not Discussed:**

N/A

**Experimental Designs Or Analyses:**

There are no experiments since the paper is theoretical.

**Methods And Evaluation Criteria:**

Not applicable since the paper is theoretical.

**Other Comments Or Suggestions:**

N/A

**Other Strengths And Weaknesses:**

N/A

**Questions For Authors:**

N/A

**Relation To Broader Scientific Literature:**

With theoretical papers like this it is difficult to assess contributions to the broader scientific literature, as they may only become apparent in the future when the results are used in a more applied context.

**Theoretical Claims:**

Regrettably, I haven't got the background to easily follow the theorems and proofs so defer to other reviewers on this.

---

> ### Author Rebuttal · Authors · 2025-03-31
>
> Thank you for your considered review.
>
> We see no questions or concerns in your review needing response. We nonetheless welcome any further discussion, e.g., as prompted by the other reviews.

---

### Official Review · Reviewer_jvgm · 2025-03-19

**Overall Recommendation:** 5

**Summary:**

This paper proposes and studies UFMAC, a unifying arithmetic circuit formalism for representing functions that support tractable marginalization. It shows that UFMACs subsume prior such tractable classes, proves that all UFMACs support polynomial time marginalization, and also shows that the reverse isn't true -- that not all funcations with tractable marginalization have UFMACs. There are some additional contributions as well.

This work substantially enhances our understanding of tractable marginalization. Additionally, the paper is carefully written (modulo some small typos in a later part) and uses neat and elegant arguments in all major proofs.

E.g., the argument that UFMACs support tractable marginalization (Prop 3.7) is refreshingly simple, leveraging an earlier observation of Juma et al (2009). Similarly, while I wasn't quite sure what the motivation for studying Hamming Weight Marginalization is, it became evident later that this tool (besides being of interest by itself) will help prove that not all tractable functions have UFMACs --- because if they have a UFMAC, then even HMAR would be tractable, which conflicts with standard complexity assumptions on certain specific problems.

I enjoyed reading the arguments, even though I am not very familiar with the area.

**Claims And Evidence:**

Yes. As noted above, the arguments are made very clearly and the proofs generally precise and elegant.

**Essential References Not Discussed:**

None to my knowledge.

**Experimental Designs Or Analyses:**

N/A

**Methods And Evaluation Criteria:**

The paper is theoretical and provides appropriate definitions, formalisms, and proofs.

**Other Comments Or Suggestions:**

Some typos to revisit:

* page 5, defn 4.1: "f \in PHM is HMAR(f) is..." => should this be "f \in PHM if FMAR(f) is..." ?

* page 8, defn 5.1: similar note as above

* page 5, 2nd line after defn 4.1: should M_{m,k} be X_{m,k}?

* page 5, line after prop 4.3: should FMACs be UFMACs? Similarly in the paragraph after lemma 4.4 and the proof of prop 4.5.

**Other Strengths And Weaknesses:**

One thing I felt the paper doesn't motivate or revisit enough (other than in the initial part of section 2.3) is the "finally" part of UFMACs. The motivation of allowing polynomial degree at the intermediate nodes is good -- but even the fact that determinants would be hard (impossible?) to cover without it is only lightly mentioned within a paragraph. Later sections don't refer to the "finally" part at all (I suppose it's not directly relevant anymore for tractability bounds? But then it must make the proofs a little bit harder / more general than assuming linearity even on internal nodes. So it feels to me that something can still be said about it.)

I wonder if even a proposition can be stated that UFMACs are strictly more expressive than UMACs.

Fig 1 is useful to have but doesn't seem to be discussed directly after the appropriate notions have been defined and/or results stated.

**Questions For Authors:**

I didn't look carefully but with a quick pass, I couldn't quite connect with section 5, virtual evidence marginalization, in part because Defn 5.1 defines a "marginalization" problem without any "summation" in it. Compare, e.g., to defns. 3.1 and 4.1, which both have a clear summation as expected. Can you clarify what exactly is the marginalization problem here? Which variables are we marginalizing over, under what setting?

**Relation To Broader Scientific Literature:**

The proposed formalism of UFMACs generalizes, to my knowledge, all prior classes for which tractability of marginalization was known.

**Theoretical Claims:**

Mostly yes, except I only lightly skimmed the proof of Theorem 4.10 (hardness of HMAR(f_aff) and the material in section 5 (virtual evidence marginalization). The findings in section 5 are a useful addition the main paper, though I think the paper is already strong even by the end of section 4.

---

> ### Author Rebuttal · Authors · 2025-04-01
>
> Thank you for your considered review. We are glad you found our work “neat,” “precise,” and “elegant,” even sometimes “refreshingly simple.” We provide answers to your questions below and welcome any further discussion.
>
> [The “finally” in UFMAC] The choice to allow polynomial degree (i.e., the “Finally” in UFMAC) has indeed been made to allow for flexible circuits like those known for the determinant. The question of separating finally multilinear circuits from multilinear circuits is currently still an open problem [1, Open Problem 3], and we agree that this can be stated more clearly in the paper and we will do so.
>
> [Virtual evidence marginalization] This is a good question; we will clarify this in the paper. The point is as follows. On the one hand, multiplying the inputs by arbitrary rationals can be viewed as updating the distribution given virtual evidence, but on the other hand, we have observed (Lemma 3.6) that marginalization of f can be reduced to evaluating the multilinear polynomial for f at 1/2,…,1/2. Therefore the evaluation of p at arbitrary rational points a_1,…,a_n can be viewed as first applying virtual evidence corresponding to 2a_1,…,2a_n to x_1,…,x_n to get 2a_1x_1,…,2a_nx_n and then evaluating at x_1=…=x_n=1/2, i.e., marginalizing. Thus there is, in a sense, an implicit summation in every evaluation of the multilinear polynomial at rational points, i.e., in every VMAR(f) instance. In particular, to observe some virtual evidence and then compute a marginal probability, a VMAR(f) query suffices.
>
> We thank you for your other notes on minor typos.
>
> [1] Shpilka and Yehudayoff. Arithmetic Circuits: a survey of recent results and open questions. Foundations and Trends in Theoretical Computer Science, 2010.

---

### Decision · Program_Chairs · 2025-05-01

**Decision:**

Accept (poster)

**Comment:**

This work studies the computational complexity of *marginalization* --- the computational problem of summing a function over al possible assignments to a subset of inputs.
It is a clean self-contained work that presents a negative result to the question: ``Can all functions with polynomial-time marginalization algorithms be expressed with polynomial-sized arithmetic circuits?''
In particular, the authors show that there exists a simple family of functions with tractable marginalization (i.e., can efficiently compute its sum over a subset of all possible inputs)
that does not have an efficient representation (i.e., polynomial-sized circuit), assuming FP != #P (which is implied by P != NP).

After the rebuttal phase, all reviewers lean towards acceptance (scores: 5, 3, 4, 3). All reviewers commented that it was a pleasure to read, but some pointed out that the connections to ICML's topics of interest seem tenuous. In the rebuttal to Reviewer WSxE, the authors presented a list of papers related to the theory of tractable marginalization that have recently been published in ICML/ICLR/NeurIPS.